# A *de novo* variant in the keratin 1 gene (*KRT1*) in a Chinese shar-pei dog with severe congenital cornification disorder and non-epidermolytic ichthyosis

Verena K. Affolter[1]*, Sarah Kiener[2,3], Vidhya Jagannathan[2,3], Terry Nagle[4], Tosso Leeb[2,3]

**1** Department of Pathology, Microbiology, Immunology, School of Veterinary Medicine, University California Davis, Davis, California, United States of America, **2** Institute of Genetics, Vetsuisse Faculty, University of Bern, Bern, Switzerland, **3** DermFocus, University of Bern, Bern, Switzerland, **4** Sacdermvet at Vista Veterinary Specialists, Sacramento, CA, United States of America

* vkaffolter@ucdavis.edu

**Data Availability Statement:** All relevant data are within the manuscript and its Supporting Information files.

## Abstract

A 3-months old Chinese shar-pei puppy with ichthyosis was investigated. The dog showed generalized scaling, alopecia and footpad lesions. Histopathological examinations demonstrated a non-epidermolytic hyperkeratosis. The parents of the affected puppy did not show any skin lesions. A trio whole genome sequencing analysis identified a heterozygous *de novo* 3 bp deletion in the *KRT1* gene in the affected dog. This variant, NM_001003392.1: c.567_569del, is predicted to delete a single asparagine from the conserved coil 1A motif within the rod domain of KRT1, NP_001003392.1:p.(Asn190del). Immunohistochemistry demonstrated normal levels of KRT1 expression in the epidermis and follicular epithelia. This might indicate that the variant possibly interferes with keratin dimerization or another function of KRT1. Missense variants affecting the homologous asparagine residue of the human KRT1 cause epidermolytic hyperkeratosis. Histologically, the investigated Chinese shar-pei showed a non-epidermolytic ichthyosis. The finding of a *de novo* variant in an excellent functional candidate gene strongly suggests that KRT1:p.Asn190del caused the ichthyosis phenotype in the affected Chinese shar-pei. To the best of our knowledge, this is the first description of a *KRT1*-related non-epidermolytic ichthyosis in domestic animals.

## Introduction

Ichthyoses are a heterogeneous group of hereditary cornification disorders. They are characterized by generalized dry skin, scaling and/or hyperkeratosis. Several genetically distinct forms have been identified in a variety of dog breeds [1]. An epidermolytic form with autosomal recessive inheritance due to a variant in epidermal keratin 10 (*KRT10*) has been documented in the Norfolk terrier [2]. Six other canine ichthyosis forms that are characterized at the molecular level represent non-epidermolytic ichthyoses. A *PNPLA1*-associated autosomal

**Funding:** This research was funded by the Swiss National Science Foundation, grant number 310030_200354 PI: Dr. Tosso Leeb, University Bern Switzerland. The funders had no role in study design, data collection and analysis, decision to publish, or preparation of the manuscript.

**Competing interests:** The authors have declared that no competing interests exist.

recessive form that involves altered glycerophospholipid metabolism has been reported in golden retrievers [3, 4]. A loss of function variant in the *TGM1* gene encoding transglutaminase 1 leads to autosomal recessive ichthyosis in Jack Russell terriers due to calcium dependent cross-linking of peptides (e.g. involucrin, loricrin) involved in forming the cornified envelope [5]. *NIPAL4* (ichtyn) deficiency was reported in ichthyotic American bulldogs [6, 7]. *ABHD5*-related autosomal recessive ichthyosis represents another defect in lipid metabolism that has been reported in golden retrievers [8]. An autosomal dominant form of ichthyosis in a German shepherd dog was caused by a missense variant in *ASPRV1* encoding a protease required for the posttranslational processing of profilaggrin [9]. Finally, an autosomal recessive *SLC27A4*-related severe syndromic form of ichthyosis has been reported in Great Danes [10, 11]. Moreover, cornification disorders suggestive of ichthyosis have been described based on clinical examination and histopathologic changes in soft-coated wheaten terriers, West Highland white terriers, English springer spaniels, Labrador retrievers [1].

While cases of *KRT1*-related ichthyosis have been documented in humans, they have not been reported in dogs to date [12–14]. This investigation documents a congenital cornification disorder in a Chinese shar-pei puppy due to a 3 base pair deletion in the *KRT1* gene.

## Materials and methods

### Clinical examinations

Clinical evaluation of the patient was performed by a board certified veterinary dermatologist (TN). Skin scrapings and skin cytology were performed, and punch biopsies from the right antebrachium, the neck and right shoulder were collected for histopathologic and immunohistochemical examination. Blood samples from patient and his parents were collected for genetic testing.

### Histopathological and immunohistochemical examinations

Submitted formalin fixed punch biopsies were bisected. Four-micron, hematoxylin and eosin-stained paraffin-sections and immunohistochemical stains were evaluated by a board certified veterinary pathologist (VKA).

Immunohistochemistry for KRT1 expression was performed on all three biopsy samples from the patient as well as on sections healthy skin from two Chinese shar-pei dogs (and other breeds (standard poodle, terrier-mix, boxer)). Four micron paraffin sections were collected on "plus" coated slides and air dried at 37˚C overnight and subsequently deparaffinized (xylene: 10 min 2x, followed by 100% ethanol: 1 min 3x, 95% ethanol: 1 min and 70% ethanol: 1min). After quenching of endogenous peroxidase (500 ul 10% sodium azide; 500 ul 30% hydrogen peroxide in 50 ml PBS; 25 min at room temperature), slides were rinsed in PBS 3x and immersed in preheated antigen retrieval solution (1x Dako Target Retrieval Solution; stock solution S1699, pH6); retrieval was performed a pressure cooker for 5 minutes. Slides were cooled down to room temperature, washed in PBS 3x. After exposing slides to 10% horse serum in PBS (15 min) the anti-CK-1 antibody (Clone 4D12B3: sc-65999; Santa Cruz Biotechnology, Inc. Dallas, Texas USA; 1:500 dilution in 10% horse serum in PBS) was applied for 60 min. After three rinses in PBS the following steps were performed: application of ImmPRESS HRP Horse Anti-Mouse IgG Polymer Reagent (Vector Cat.# MP-7402; 30 min), thorough PBS rinses and addition of substrate (Vector, SK-4800). Development was monitored microscopically and reaction was stopped by immersing the slides in Milli-Q/distilled water. Counterstain (Gill's Hematoxylin #2 RICCA, 3536–16; 15–30 s) was stopped by washing slides in running tap water. Slides were then cover-slipped using Shandon-Mount media (Thermo Scientific, 1900331).

## Genetic examinations

**Animal selection.** This study included a total of 22 Chinese shar-peis. They comprised one ichthyosis affected Chinese shar-pei and its unaffected parents. Additional samples from 19 unrelated Chinese shar-peis without clinical signs of ichthyosis from the Vetsuisse Biobank were used as controls.

**Whole-genome sequencing.** Illumina TruSeq PCR-free libraries with insert sizes of ~330 bp were prepared from the affected dog and both parents. The libraries were sequenced with 2 x 150 bp chemistry on a NovaSeq 6000 instrument. The reads were mapped to the CamFam3.1 reference genome assembly as described [15]. The sequence data were submitted to the European Nucleotide Archive with the study accession PRJEB16012 and sample accessions SAMEA7198604 (affected puppy), SAMEA7198605 (unaffected dam) and SAMEA7198612 (unaffected sire). Variant calling was performed as described [15]. To predict the functional effects of the called variants, the SnpEff software [16] together with NCBI annotation release 105 for the CanFam 3.1 genome reference assembly was used. For variant filtering, we used 793 control genomes derived from 784 dogs and 9 wolves (S1 Table). We applied two different hard filtering approaches for homozygous and heterozygous private variants in the affected dog: In the search for private homozygous variants, we retained only variants with genotype 1/1 in the affected puppy and genotypes 0/0 or ./. in the 793 control genomes. In the search for private heterozygous variants, we retained only variants with genotype 0/1 in the affected puppy and genotypes 0/0 or ./. in the 793 control genomes. Subsequently, the private variants were combined in an Excel-file for further inspection (S2 Table). For functional prioritization, variants with SnpEff impact predictions high or moderate were combined and termed "protein-changing variants".

**Confirmation of parentage.** To confirm the parentage of the presumed parents and the affected dog, we used the genome sequence data (vcf-file) of the affected dog and its parents. Using PLINK v1.9 we extracted 6,269,532 informative markers distributed over all autosomes and performed a pairwise IBD estimation with the—genome command [17]. The sire-offspring and dam-offspring pairs both had an estimated overall IBD proportion (PI_HAT) of 50% with 0% P(IBD = 0), 100% p(IBD = 1) and 0% p(IBD = 2) as expected for parent-offspring duos.

**Gene analysis.** We used the dog reference genome assembly CanFam3.1 and NCBI annotation release 105. Numbering within the canine *KRT1* gene corresponds to the NCBI RefSeq accession numbers NM_001003392.1 (mRNA) and NP_001003392.1 (protein). For a multiple species comparison of KRT1 amino acid sequences, we used these accessions: NP_006112.3 (*Homo sapiens*), NP_001104288.1 (*Pan troglodytes*), XP_002687292.1 (*Bos taurus*), NP_032499.2 (*Mus musculus*), NP_001008802.2 (*Rattus norvegicus*). A precomputed multiple species sequence alignment was obtained from the NCBI HomoloGene website (https://www.ncbi.nlm.nih.gov/homologene).

**Sanger sequencing.** We used Sanger sequencing to confirm the *KRT1*:c.567_569del variant and to perform targeted genotyping of all samples. AmpliTaqGold360Mastermix (Thermo Fisher Scientific, Waltham, MA, USA) and the primers 5'-CCT GGT GGC ATA CAG GAA GT-3' (forward primer) and 5'-CTC GTT CGC ACC CTA GAA AG-3' (reverse primer) were used to amplify a 454 bp product. After treatment with shrimp alkaline phosphatase and exonuclease I, PCR amplicons were sequenced on an ABI 3730 DNA Analyzer (Thermo Fisher Scientific). Sanger sequences were analyzed using the Sequencher 5.1 software (GeneCodes, Ann Arbor, MI, USA).

## Ethics statement

All animal experiments were performed according to the local regulations. The dogs in this study were privately owned and skin biopsies and blood samples for diagnostic purposes were

collected with the consent of the owner. The collection of blood samples from healthy dogs was approved by the "Cantonal Committee for Animal Experiments" (Canton of Bern, Switzerland; permit 71/19).

## Results

### Family anamnesis, clinical examinations, histopathology

A 3-months old male Chinese shar-pei was presented for scaly skin and reduced overall body growth when compared with his 3 littermates, a female and 2 male puppies with clinically normal skin. Dam and sire were in the same household and clinically normal. Administration of Clavamox (ZoetisUS; 62.5 mg twice daily for 10 days) followed by Convenia (ZoetisUS; 80 mg/ml 0.54 ml), prednisone (5 mg twice daily, then once daily) and frequent bathing with Hexa-Chlor-K shampoo (GelnHaven Therapeutics, Schuyler, Oregon) revealed minimal improvement. Terramycin eye ointment (ZoetisUS) had been applied for entropium of the left eye. At time of presentation, the dog appeared bloated and uncomfortable despite eating and consuming normal amounts of water.

At the time of presentation severe generalized scaling and alopecia was noted, with scaling most prominent on the head (Fig 1A), neck (Fig 1B), abdomen (Fig 1C), legs, axillary folds (Fig 1D) and paws. Prominent follicular fronds accompanied surface scaling. The paw pads appeared deformed and hyperkeratotic. Pruritus was not observed. The left eye had an entropium. Skin scrapings for Demodex mites were negative. Skin cytology revealed numerous yeast organisms.

Biopsies from all three locations revealed severe hyperkeratosis, characterized by prominent keratin lamellae overlaying a marked compact layer of keratin (Fig 2A and 2B). The epidermis was markedly acanthotic and most infundibular regions were markedly dilated resulting in narrowing of the interfollicular epidermis (Fig 2A). The follicular lumina were filled with keratin and the infundibular epithelium was hyperplastic. Some perinuclear clearing was most evident in the prominent granular layer with irregularly sized keratohyalin (Fig 2C). Dispersed mast cells and some plasma cells and neutrophils were present in the superficial dermis and the sebaceous glands were prominent. Several small neutrophilic crusts with some cocci were noted entrapped within the thick keratin layer (Fig 2D). In the sample from the shoulder some follicles contained neutrophils in their lumina and the epidermis was covered by parakeratosis. Superficial yeast organisms were not observed in sections stained with periodic acid-Schiff stain. Many hair follicles and remaining hair shafts contained clumped melanin. The following morphologic diagnoses were made: 1) severe acanthosis and superficial and follicular hyperkeratosis suggestive of a cornification disturbance and 2) multifocal neutrophilic pustular dermatitis and neutrophilic luminal folliculitis and 3) melanin pigment clumping indicating dilute hair coat color. The latter was considered an expected incidental finding as the dog had a $d^1/d^2$ genotype at the *MLPH* gene [18, 19] and was born out of two clinically inconspicuous dilute-colored parents.

Given the overwhelming features of follicular and superficial hyperkeratosis, a hereditary cornification disorder consistent with ichthyosis was considered. Pustules and superficial folliculitis indicated a secondary pyoderma, which, based on skin cytology, was accompanied by a superficial yeast infection.

### Genetic analysis

In order to characterize the underlying causative genetic variant we sequenced the genome of the affected puppy at 18.9x coverage and searched for variants in 36 candidate genes for

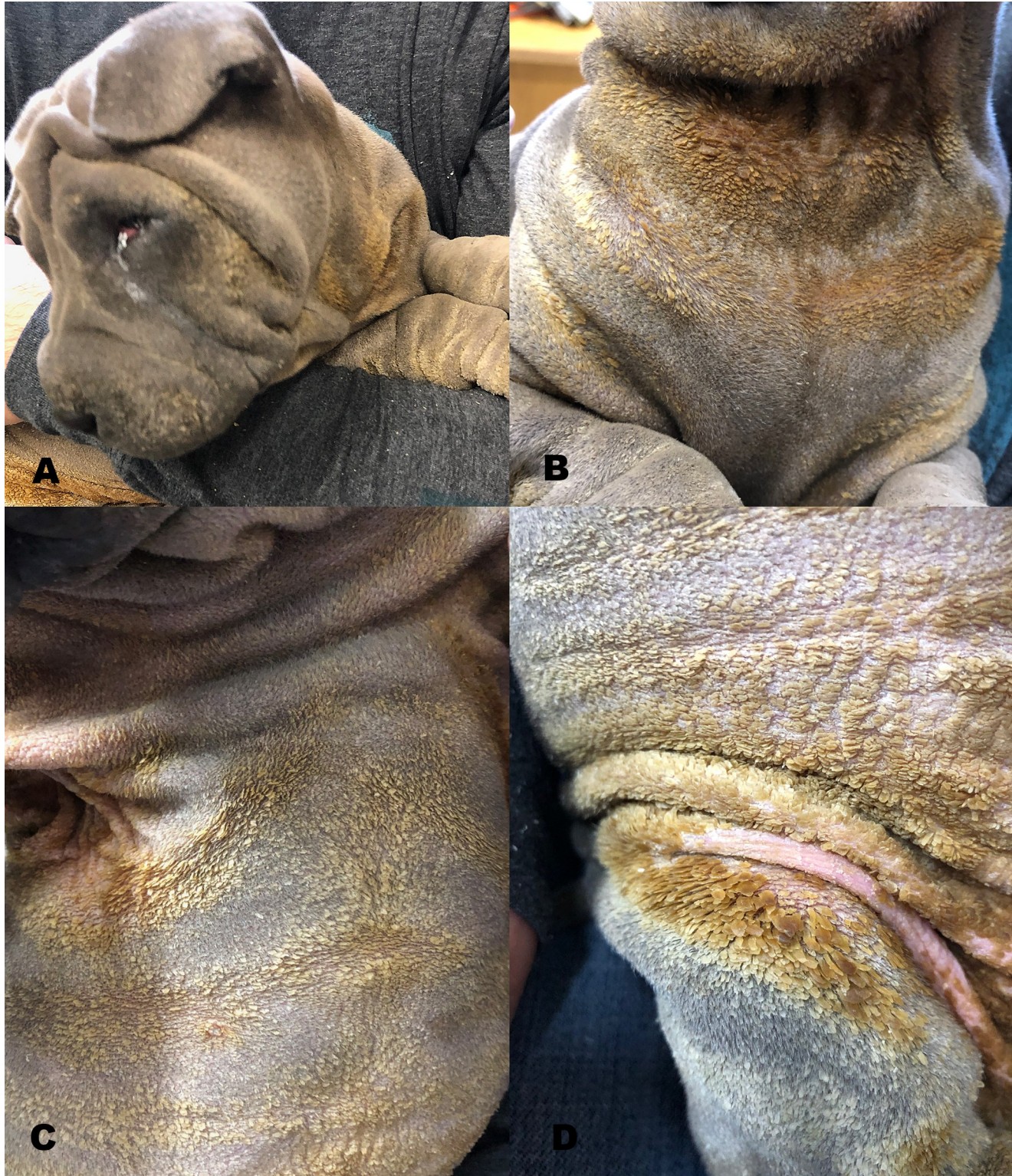

**Fig 1. Clinical presentation of affected Chinese shar-pei.** Severe generalized alopecia and scaling with marked follicular fronds on (A) head, (B) neck, (C) abdomen and (D) axillary folds.

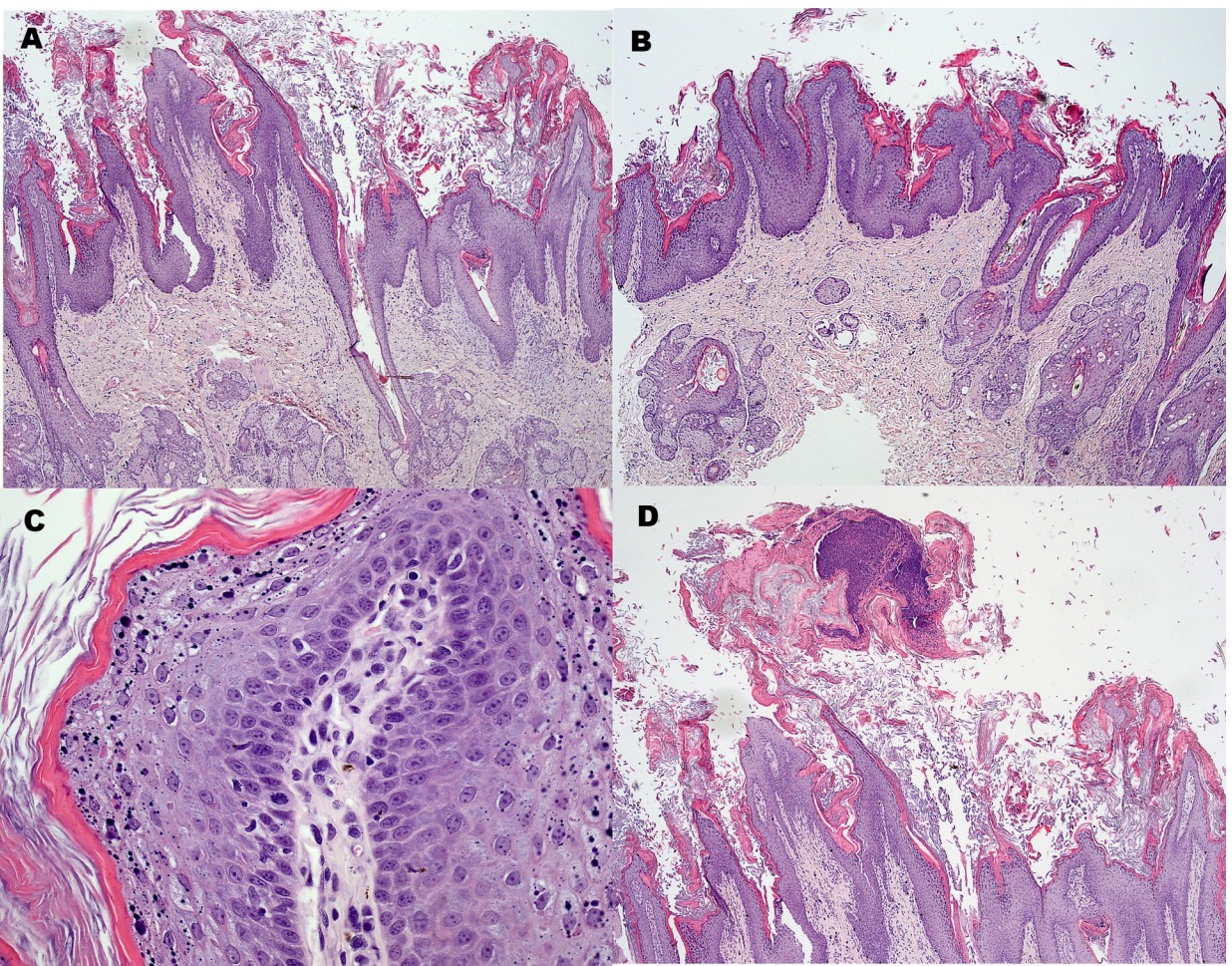

**Fig 2. Histopathologic changes in the affected Chinese shar-pei.** (A and B) The epidermis is severely hyperplastic with extensive compact and lamellar hyperkeratosis of the surface. The hyperkeratosis extends into the follicular lumina dilating the follicular openings (H&E; 40x). (C) There is a prominent granular layer with irregularly sized keratohyalin granules and common perinuclear vacuoles (H&E; 400x). (D) Multifocally, there were neutrophilic crusts indicating secondary pyoderma.

ichthyosis (S2 Table), which were exclusively present in the affected dog and absent from 793 control genomes (Tables 1 and S1 and S3).

Subsequently, we performed a trio analysis and compared the genotypes in the affected dog with the genotypes of both parents (S3 Table). We considered two alternative scenarios for the putative causal variant: For an autosomal recessive trait, we expected the affected dog to be homozygous for the alternate allele and both parents heterozygous. Alternatively, for a

**Table 1. Variants detected by whole genome sequencing of the affected Chinese shar-pei.**

| Filtering step | heterozygous variants | homozygous variants |
|---|---|---|
| Variants in the whole genome | 4,261,447 | 2,920,513 |
| Private variants[a] | 82,046 | 9,522 |
| Private protein-changing variants[a] | 503 | 39 |
| Private protein-changing variants in 36 candidate genes[a] | 2 | 0 |

[a]The parents of the affected dog were excluded for these filtering steps.

dominant trait that could only have been caused by a *de novo* mutation event, the affected dog should be heterozygous and both parents should be homozygous for the reference allele. The results of the trio analysis are summarized in Table 2.

Taken together these analyses identified a single protein-changing variant in a known ichthyosis candidate gene, for which the genotypes of the parents were compatible with a pathogenic effect. The variant was a heterozygous in frame deletion in the first exon of *KRT1* (NM_001003392.1:c.567_569del) (Fig 3A), removing three nucleotides coding for an asparagine of the 1A coil domain (NP_001003392.1:p.(Asn190del), Fig 3B). The formal genomic designation of the variant is Chr27:2,422,716_2,422,718del (CanFam3.1).

The trio analysis comparing the variants in the affected dog with the genomes of both parents revealed that *KRT1*:c.567_569del represented a *de novo* variant as the mutant allele was absent from leukocyte DNA of both parents. The correct parentage of sire and dam in this family was confirmed based on the genome sequence data.

We used Sanger sequencing to confirm the identified candidate *KRT1*: c.567_569del variant and to genotype the rest of the Chinese shar-peis from our study. The deletion was only present in heterozygous state in the ichthyosis affected puppy whereas both parents and all remaining Chinese shar-peis were homozygous for the wild type allele.

## Expression of KRT1

Given the *de novo* variant in the *KRT1* gene in this Chinese shar-pei, KRT1 expression of the tissue was evaluated by light microscopy using immunohistochemistry. The intensity of KRT1 expression in the epidermis and the follicular epithelia of the affected dog was visually comparable to normal skin samples from Chinese shar-pei dogs (Fig 4) and other breeds. Keratinocytes revealed strong membranous and cytoplasmic KRT1 expression.

## Discussion

Reported canine hereditary cornification disorders are due to genetic variants affecting either keratins or components involved in crosslinking of peptides or disruption of lipids within the cornified envelope [1]. All of them lead to interruption of successful cornification or desquamation. Most well documented forms of canine ichthyosis are non-epidermolytic, involving genetic variants in *ABHD5* or *PNPLA1* in golden retrievers, *TGM1* in Jack Russell terriers, *NIPAL4* in American bulldogs, and *ASPRV1* in a German shepherd dog [3–11, 21].

Marked follicular fronds in addition to prominent surface scales observed in this Chinese shar-pei indicated a cornification disturbance affecting both epidermis as well as follicles. Follicular fronds appear to be less prominent in other breeds with non-epidermolytic ichthyosis [3, 5, 6, 21].

Associated with the marked epidermal and follicular hyperkeratosis was a rather prominent acanthosis. The latter was also present in areas with no evidence of secondary pyoderma. This is somewhat in contrast with non-epidermolytic ichthyosis in other dog breeds, where marked

**Table 2. Trio analysis of the affected Chinese shar-pei and its parents.**

| Filtering step | heterozygous variants | homozygous variants |
|---|---|---|
| Private variants that were absent from 793 control genomes | 82,046 | 9,522 |
| Protein-changing & genotypes of parents compatible with a pathogenic effect | 27 | 19 |
| Protein-changing & genotypes of parents compatible & in 36 candidate genes | 1 | 0 |

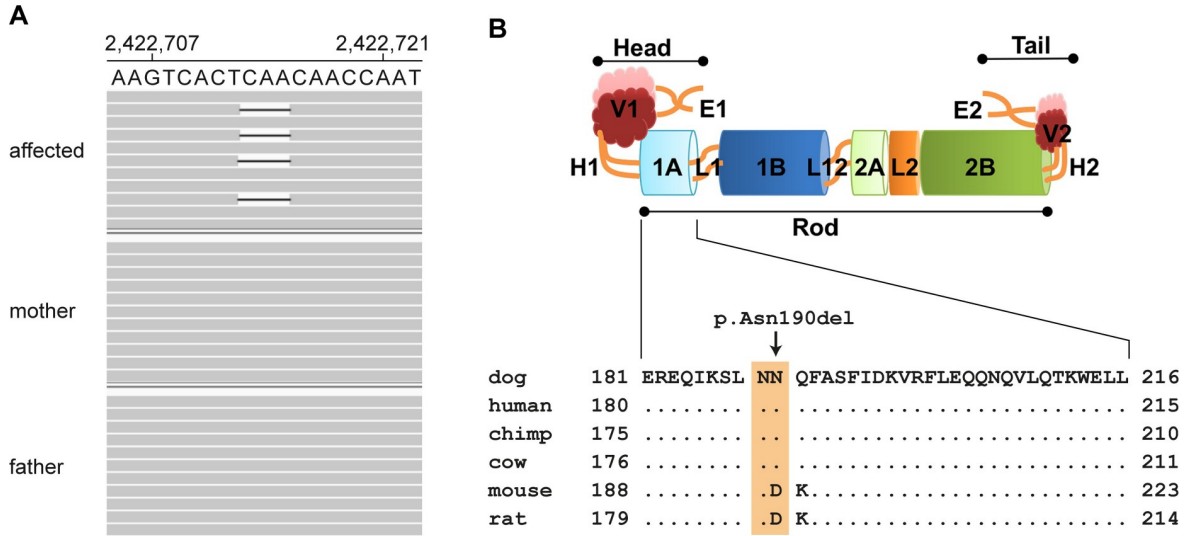

**Fig 3. Details of the *KRT1*:c.567_569del, p.(Asn190del) variant.** (A) Integrative Genomics Viewer (IGV) screenshot showing the short-read alignments of the ichthyosis affected puppy and its non-affected parents at the position of the deletion. A deletion of one copy of the allele is visible in the case but not in the parents. Note that in the IGV screenshot bases 2,422,713–2,422,715 are deleted, whereas the 3'-rule of HGVS nomenclature requires to designate this variant as Chr27:2,422,716_2,422,718del (CanFam3.1). (B) Schematic representation of the protein domain structure of a keratin dimer [20] with the highly conserved amino acid sequence of the coil 1A subdomain shown below. The variant is predicted to delete an asparagine residue from coil 1A, which is located within the rod domain of KRT1.

hyperkeratosis is often disproportionate to the degree of epidermal acanthosis [5, 6, 21]. However, the marked hypergranulosis with irregularly sized keratohyalin granules as well as the presence of mild perinuclear vacuolization of keratinocytes was a finding consistent with features seen in other breeds. Electron microscopy revealed curvilinear membranous material within the granular layer cytosol, also in particular in the perinuclear swellings seen on light microscopy [6]. Electron microscopic evaluation was not performed in the Chinese shar-pei investigated by us.

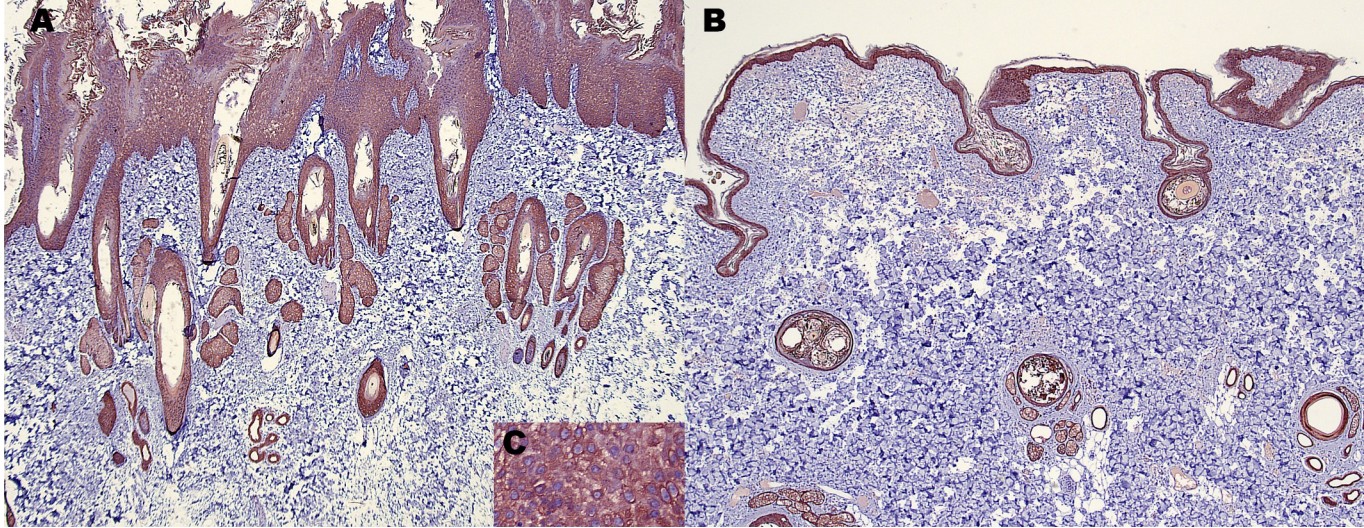

**Fig 4. KRT1 expression in Chinese shar-pei skin (immunohistochemistry with anti-KRT1 antibody).** (A) Intensity of KRT1 expression in the epidermis and follicular epithelia of the affected Chinese shar-pei is comparable to (B) normal Chinese shar-pei skin (40x). (C) Strong KRT1 expression in the epidermis of the affected Chinese shar-pei (400x).

Genetic variants in *KRT1* or *KRT10* in humans typically present with an epidermolytic ichthyosis [12, 22–24]. The *KRT1*-related epidermolytic hyperkeratosis presents with or without palmo-plantar keratoderma, while *KRT10*-related epidermolytic ichthyosis typically does not involve palmo-plantar keratoderma. A *KRT10* variant in Norfolk terriers results in epidermolytic ichthyosis [2], but the *KRT1* variant in the Chinese shar-pei of this investigation was not associated with epidermolytic changes. Altered KRT1/KRT10 dimer formation due to variants affecting the structure of the paired 2B and V2 domains leads to severe acanthosis and hyperkeratosis in humans [25], reflecting the histologic features observed in the affected Chinese shar-pei. In human literature this is also referred to as "epidermolytic ichthyosis sine epidermolysis" [26].

The identified variant in the affected Chinese shar-pei leads to a deletion of 3 bases coding for an asparagine in the coil 1A motif of KRT1, which is part of the central rod domain of the KRT1/KRT10 heterodimer. The sequence of the central rod domain is highly conserved amongst epidermal keratins [20]. Three different missense variants affecting the homologous asparagine-188 of KRT1 in human patients have previously been described to cause epidermolytic hyperkeratosis [22, 27, 28]. This strongly suggests that changes in this region are intolerable for functional keratin filament formation.

In the affected Chinese shar-pei, the mutant KRT1 protein lacking asparagine-190 is apparently expressed within the acantholytic epidermis and the follicular epithelia as demonstrated by the immunohistochemistry experiment. The expression level was comparable to control skin tissue from an unaffected Chinese shar-pei and dogs from other breeds. This might indicate that the single amino acid deletion possibly interferes with dimerization or another function of the KRT1 molecule.

Our genetic analysis revealed *KRT*:p.(Asn190del) as a highly plausible candidate causal variant for the observed phenotype. According to the ACMG/AMP consensus standards for the interpretation of sequence variants in human patients [29], our data provide one strong, three moderate and one supporting criteria for pathogenicity, which is sufficient to classify the variant as pathogenic. The strong criterion is the demonstration of a *de novo* variant in an affected dog born out of two healthy parents. The three moderate criteria are the absence of the mutant allele from a relatively large control cohort, the protein length change due to an in-frame deletion and the fact that missense variants affecting the same amino acid have been established as pathogenic in humans. We consider the highly specific disease phenotype with known monogenetic etiology as supporting criterion for pathogenicity. Although our analysis yielded a variant fulfilling diagnostic criteria for pathogenicity, we have to caution that we did not investigate structural variants. Furthermore, the analysis relied on the accuracy and completeness of the CanFam3.1 genome assembly and NCBI annotation release 105. Therefore, we cannot formally exclude the possibility that other potentially plausible variants were missed. NCBI annotation release 105 agrees with a manually curated annotation of the canine *KRT1* gene [30].

In summary, to the best of our knowledge, this is the first case of a *KRT1*-related ichthyosis reported in domestic animals and the first case of ichthyosis in a Chinese shar-pei dog.

## Supporting information

**S1 Table. Accession numbers of 796 dog/wolf genome sequences.** The affected dog is highlighted in red and the non-affected parents are highlighted in blue. Both parents and the affected dog were used for a trio analysis. The other 793 genome sequences were used as controls in filtering for private variants.
(XLSX)

**S2 Table. 36 candidate genes for ichthyosis.** List of non-syndromic and syndromic forms of ichthyosis.
(XLSX)

**S3 Table. Private variants in the affected Chinese shar-pei.** Variants are listed multiple times, if they have predicted effects on more than one transcript. Variants with a SnpEff predicted impact of "high" or "moderate" were considered protein-changing variants. For private variant filtering, the genomes of the case and 793 controls (excluding the parents) were considered. The KRT1:c.567_569del variant is highlighted in yellow (line 176,297).
(XLSX)

## Acknowledgments

The authors are grateful to all dog owners who donated samples and shared health and pedigree information of their dogs. We thank Nathalie Besuchet Schmutz, Catia Coito, Marion Ernst and Daniela Steiner for expert technical assistance, the Next Generation Sequencing Platform of the University of Bern for performing the high-throughput sequencing experiments, and the Interfaculty Bioinformatics Unit of the University of Bern for providing high performance computing infrastructure. We also thank Kristy Harmon of the Leukocyte Antigen Biology Laboratory at University California Davis for establishing the immunohistochemistry with the anti-cytokeratin 1 antibody. We thank the Dog Biomedical Variant Database Consortium (Gus Aguirre, Catherine André, Danika Bannasch, Doreen Becker, Brian Davis, Cord Drögemüller, Kari Ekenstedt, Kiterie Faller, Oliver Forman, Steve Friedenberg, Eva Furrow, Urs Giger, Christophe Hitte, Marjo Hytönen, Vidhya Jagannathan, Tosso Leeb, Frode Lingaas, Hannes Lohi, Cathryn Mellersh, Jim Mickelson, Leonardo Murgiano, Anita Oberbauer, Sheila Schmutz, Jeffrey Schoenebeck, Kim Summers, Frank van Steenbeek, Claire Wade) for sharing whole genome sequencing data from control dogs. We also acknowledge all researchers who deposited dog or wolf whole genome sequencing data into public databases.

## Author Contributions

**Conceptualization:** Verena K. Affolter.

**Data curation:** Verena K. Affolter, Sarah Kiener, Vidhya Jagannathan, Terry Nagle, Tosso Leeb.

**Investigation:** Verena K. Affolter, Tosso Leeb.

**Project administration:** Verena K. Affolter.

**Resources:** Verena K. Affolter.

**Supervision:** Verena K. Affolter.

**Writing – original draft:** Verena K. Affolter, Tosso Leeb.

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
