## [Decision Letter · Decision Letter 0]

12 Apr 2022

PONE-D-21-30440

A de novo variant in the keratin 1 gene (KRT1) in a Shar Pei dog with severe congenital cornification disorder and non-epidermolytic ichthyosis

PLOS ONE

Dear Dr. Affolter,

Thank you for submitting your manuscript to PLOS ONE. After careful consideration, we feel that it has merit but does not fully meet PLOS ONE’s publication criteria as it currently stands. Therefore, we invite you to submit a revised version of the manuscript that addresses the points raised during the review process.

Editor and reviewers find the manuscript to be largely sound and worthy of publication, however numerous inconsistencies in language and clarity need to be addressed as detailed in the reviewer comments.

We look forward to receiving your revised manuscript.

Kind regards,

Brian W Davis

Academic Editor

PLOS ONE

“This research was funded by the Swiss National Science Foundation, grant number 310030_200354 PI: Dr. Tosso Leeb, University Bern Switzerland”

Reviewers' comments:

Reviewer's Responses to Questions

**Comments to the Author**

1. Is the manuscript technically sound, and do the data support the conclusions?

Reviewer #1: Yes

Reviewer #2: Yes

2. Has the statistical analysis been performed appropriately and rigorously? 

Reviewer #1: Yes

Reviewer #2: N/A

3. Have the authors made all data underlying the findings in their manuscript fully available?

Reviewer #1: Yes

Reviewer #2: Yes

4. Is the manuscript presented in an intelligible fashion and written in standard English?

Reviewer #1: Yes

Reviewer #2: Yes

5. Review Comments to the Author

Reviewer #1: The manuscript entitled “A de novo variant in the keratin 1 gene (KRT1) in a Shar Pei dog with severe congenital cornification disorder and non-epidermolytic ichthyosis” is a case study that describes the identification of a putative causal genetic variant in a single dog. While the identified variant is deemed to be de novo, and is not expected to have broad clinical relevance or importance in carrier identification within vulnerable populations, the discovery does expand the knowledge base of congenital epidermal disorders and gene function.

Minor corrections

- Throughout, a consistent scheme for capitalization of dog breed names needs to be followed. The generally accepted convention is to capitalize only the proper nouns in the name. Likewise, the official breed names should be used when relevant.

o Throughout: Shar Pei = Chinese shar pei

o Pg 3, line 49: Golden retriever = golden retriever

o Pg 3, line 54-55: German shepherd = German shepherd dog

o Pg 3, line 58: soft-coated Wheaten terriers = soft-coated wheaten terriers

o Pg 3, line 59: West Highland White terriers = West Highland white terriers

o Pg 3, line 59: English Springer spaniels = English springer spaniels

o Pg 4, line 78: Standard Poodle = standard poodle

o Pg 4, line 78: Terrier-mix = terrier mix

o Pg 4, line 78: Boxer = boxer

o Pg 10, line 244: German shepherd = German shepherd dog

o Pg 11, line 264: Norfolk Terriers = Norfolk terriers

- Pg 3, line 61: people = humans

- Pg 3, line 67: board-certified dermatologist = board certified veterinary dermatologist

- Pg 4, line 74: board-certified pathologist = board-certified veterinary pathologist

- Pg 4, line 77: “… three biopsy samples of the patient …” = “… three biopsy samples from the patient …”

- Pg 5, line 112: “… 793 control genomes (S1 Table).” Tell us roughly what the control genomes are. “… consisting of ### domestic dogs and ### wild canids.”

- Pg 6, ine 145: “… resented for scaly skin and reduced growth when …” Growth of what? Hair, skin, body size?

- Pg 7, lines 151 – 152: should read “At the time of presentation, the dog appeared …”

- Pg 7, line 159: “… for Demodex mites was negative.” Should be “… for Demodex mites were negative.”

- Pg 10, line 241: What is the envelope?

- Pg 10, line 241: Add a period after “desquamation”.

- Pg 10, line 257: “Electron microscopic revealed …”. Do you mean “electron microscopy” or is there a missing word after “microscopic”?

- Pg 11, line 259: Remove the “; it” at the end of the line.

- Pg 11, line 261: people = humans

- Pg 11, line 268: Should read “… reflecting the histologic features observed in the affected Chinese shar pei.”

- Pg 12, lines 286 – 288: The last sentence does not belong here. Move it up somewhere into the body of the Discussion.

Moderate to substantial corrections

- In the introduction, add 1 – 2 sentences describing “hereditary cornification disorders” and “ichthyosis”.

- Pg 3, lines 47 – 57: This is a single sentence that runs nearly 11 lines. It needs to be restructured.

- Pg 8, line 180 mentions that dilute hair coat color was considered incidental to the skin findings. None of the candidate genes investigated later were related to coat color or associated hair/skin conditions. Should any consideration be given to the possible overlap with color dilution alopecia, which is commonly seen with the “D locus” dilute coloration?

- Why were only known ichthyosis candidate genes considered? In the trio analysis, you identified 542 protein-changing variants, which is a very reasonable number of genes to investigate for possible functional implications. You may wind up with the same answer, I think you have sufficient evidence to suspect that the KRT1 variant is correct, but by not limiting yourself to a list of candidate genes, you are better able to make your case. What other genes were affected by the 542 protein-changing variants, and would any of them be potential functional candidates?

- Table 1: What does “Genotypes of parents compatible with a pathogenic effect” mean? It sounds like there is one variant found in the parents that may be pathogenic? What variant is that, what gene?

Reviewer #2: The authors have produced an interesting manuscript identifying a putatively associated variant in a Shar Pei with ichthyosis. The 3 base pair heterozygous deletion was uncovered by selecting private variants against a large set of public genomes, filtering with known candidate genes, and compatibility with parental genotypes. Immunohistochemistry indicated the deletion does not alter protein expression. Based on these observations and previous literature, the authors suggest the deletion may interfere with protein interaction of a known partner (or other function) and is the presumed cause of disease.

Comments

• While this does appear to be a good candidate variant, there are numerous instances where the language describing the potential role is not appropriate based on the analyses yet completed. For example lines 34-35 (“…strongly suggests…caused”), line 63 (“…due to a 3 base pair…”), and line 276 (“This strongly suggest…”). The authors should rephrase these (and any other) passages to reflect potential association as opposed to causation.

• Line 31 (and the discussion) speculate that the deletion “interferes with keratin dimerization”. While this certainly could be true based on the previous literature, this has not been thoroughly investigated here. The immunohistochemistry presented herein does not suggest this really. If the authors want to include a description of this potential mechanism the language should be toned down (as in the first comment) and likely removed from the abstract. Alternatively, and if possible, it would be really interesting to see the results from some type of in-vitro protein-protein interaction assay.

• How was the variant filtering accomplished with the 793 public control genomes? In addition, I noticed there were no Shar Pei’s in the control set. Were there none available or were they excluded? A reference (14) was used to cover how the mapping and variant calling were conducted but additional descriptions of how the variant filtering was conducted, why these 793 were selected, and why no other healthy Shar Pei’s were included is needed.

• Why was confirmation of parentage done with VCFtools as opposed to a more commonly used parentage/genetic marker test? In this section they are referred to as “presumed parents” but “unaffected parents” earlier suggesting not presumed. Similarly, lines 211-212 state parentage was “confirmed based on the genome sequence data.” Please include clarification as to the why and how here – this is important so that we have absolute confidence that the parents are in fact the parents.

• Line 122 the authors should include a description of the multiple sequence alignment method.

• The candidate gene set appears to be from three previous papers, one of which is for a dog with ichthyosis. Were there attempts to expand the search for candidate genes via databases such as OMIM or PubMed? Additional descriptions on why only these sources were used should be included. Perhaps a structured systematic review is not needed but a more complete explanation on how the candidate gene set was developed would be helpful in understanding why this severe filtering (92k to 500 variants) was needed.

• Table 1 lists 92k private variants, were there any attempts to examine this set for other potential variants compatible with an autosomal recessive inheritance pattern that could explain the phenotype? The definition of “protein-changing” should be probably be included somewhere as well. I would suggest including some type of analyses on the 92k private variants. For example, theses could be categorized by location, type, and predicted impact (I believe SnpEff outputs summary files containing this type of data?). Similar to the candidate gene comment above, the authors should examine at least the predicted high-impact variants for other potential genes that may be associated with keratinization disorder.

• Lines 231 and 235 state expression was “comparable” between the affected and normal dogs. It looks a bit a subjective and to the untrained eye, these appear rather different on paper. It would seem that in order to make statements about comparability, protein quantification should be conducted. Transcript quantification via RNA-seq or qPCR may be warranted as well to uncover any potential differences in expression. Baring any additional experiments, perhaps there is a pathologist associated with this study that could describe how these were identified as comparable?

• There should probably be some mention in the discussion of the uncertainty inherent in using the public annotation as there are known issues. For example, does this deletion appear in the same amino acid and domain using the Ensembl annotation which lists two isoforms?

Minor comments

• There is a discrepancy in author order between the cover sheet and manuscript. Please correct.

• Lines 46-57 is a single sentence and as a result a little difficult to follow. I would suggest breaking this up or restructuring it.

• Table 1 legend “genome resequencing”. Is this supposed to be “genome sequencing” or was this dog previously sequenced?

• Figure 3 legend has a reference error {Bray, 2015 #17}.

• Minor inconsistency in naming the protein dimer (line - 266 KRT1/10; line 273 - KRT1/KRT10). I am not sure which is correct but should be consistent I would think.

6. PLOS authors have the option to publish the peer review history of their article (what does this mean?). If published, this will include your full peer review and any attached files.

Reviewer #1: No

Reviewer #2: No

---

## [Author Response · Author response to Decision Letter 0]

4 Jun 2022

Responses to Reviewers:

„A de novo variant in the keratin 1 gene (KRT1) in a Chinese shar pei dog with severe congenital cornification disorder and non-epidermolytic ichthyosis”.

PONE-D-21-30440

We would like to thank the reviewers for the insightful suggestions. Please find our responses and comments listed below.

Sincerely,

Verena K. Affolter

Dr.med.vet., Dipl. ECVP, PhD

Professor of Clinical Dermatopathology 

Chief of Service Anatomic Pathology

phone: (530) 754 0104; fax: (530) 752 3349

vkaffolter@ucdavis.edu

Reviewer 1:

(1)

Minor corrections

- Throughout, a consistent scheme for capitalization of dog breed names needs to be followed. The generally accepted convention is to capitalize only the proper nouns in the name. Likewise, the official breed names should be used when relevant.

o Throughout: Shar Pei = Chinese shar pei

Response: this has been corrected throughout the document

o Pg 3, line 49: Golden retriever = golden retriever

o Pg 3, line 54-55: German shepherd = German shepherd dog

o Pg 3, line 58: soft-coated Wheaten terriers = soft-coated wheaten terriers

o Pg 3, line 59: West Highland White terriers = West Highland white terriers

o Pg 3, line 59: English Springer spaniels = English springer spaniels

o Pg 4, line 78: Standard Poodle = standard poodle

o Pg 4, line 78: Terrier-mix = terrier mix

o Pg 4, line 78: Boxer = boxer

o Pg 10, line 244: German shepherd = German shepherd dog

o Pg 11, line 264: Norfolk Terriers = Norfolk terriers

Response: these have been corrected throughout the document

- Pg 3, line 61: people = humans Response: corrected; line 76

- Pg 3, line 67: board-certified dermatologist = board certified veterinary dermatologist Response: corrected; line 107

- Pg 4, line 74: board-certified pathologist = board-certified veterinary pathologist 

Response: corrected; line 116

- Pg 4, line 77: “… three biopsy samples of the patient …” = “… three biopsy samples from the patient …” Response: corrected

- Pg 5, line 112: “… 793 control genomes (S1 Table).” Tell us roughly what the control genomes are. “… consisting of ### domestic dogs and ### wild canids.” 

Response: text gas been adjusted; lines 162-170

- Pg 6, line 145: “… presented for scaly skin and reduced growth when …” Growth of what? Hair, skin, body size? Response: adjusted

- Pg 7, lines 151 – 152: should read “At the time of presentation, the dog appeared …” Response� corrected

- Pg 7, line 159: “… for Demodex mites was negative.” Should be “… for Demodex mites were negative.” Response: corrected

- Pg 10, line 241: What is the envelope? Response: corrected to “cornified envelope”

- Pg 10, line 241: Add a period after “desquamation”. Response: corrected

- Pg 10, line 257: “Electron microscopic revealed …”. Do you mean “electron microscopy” or is there a missing word after “microscopic”? Response: corrected

- Pg 11, line 259: Remove the “; it” at the end of the line. Response� corrected

- Pg 11, line 261: people = humans. Response: corrected

- Pg 11, line 268: Should read “… reflecting the histologic features observed in the affected Chinese shar pei.” Response: corrected

- Pg 12, lines 286 – 288: The last sentence does not belong here. Move it up somewhere into the body of the Discussion. Response: changes has been made

(2)

Moderate to substantial corrections

In the introduction, add 1 – 2 sentences describing “hereditary cornification disorders” and “ichthyosis”.

Response: Revised accordingly

(3)

Pg 8, line 180 mentions that dilute hair coat color was considered incidental to the skin findings. None of the candidate genes investigated later were related to coat color or associated hair/skin conditions. Should any consideration be given to the possible overlap with color dilution alopecia, which is commonly seen with the “D locus” dilute coloration?

Response: We thank the reviewer for this important comment. Both parents of the affected puppy were also dilute-colored and did not show clinically manifest color dilution alopecia. Color dilution in various dog breeds presents with very mild follicular hyperkeratosis and may or may not be associated with alopecia. Moreover, alopecia due to color dilution is a gradually developing process, typically presenting clinical signs over a period of years. We therefore think that any potential contribution of the coat color dilution to the phenotype in the ichthyotic puppy is minimal at most. We added a short statement about the parents to this section.

(4)

Why were only known ichthyosis candidate genes considered? In the trio analysis, you identified 542 protein-changing variants, which is a very reasonable number of genes to investigate for possible functional implications. You may wind up with the same answer, I think you have sufficient evidence to suspect that the KRT1 variant is correct, but by not limiting yourself to a list of candidate genes, you are better able to make your case. What other genes were affected by the 542 protein-changing variants, and would any of them be potential functional candidates?

Response: We revised the results section and give a more detailed description of the results that hopefully better illustrates why our analyses ended up at a single candidate variant. We also revised the S3 table and give now all 91,568 private variants to allow the interested reader to evaluate other candidate genes and/or variants. We are afraid that due to space considerations, we don't think that a verbose discussion of the other genes and variants would be helpful.

(5)

- Table 1: What does “Genotypes of parents compatible with a pathogenic effect” mean? It sounds like there is one variant found in the parents that may be pathogenic? What variant is that, what gene?

Response: We agree with the reviewer that our initial presentation was confusing. We think that we have addressed this with the changes outlined under the previous comment.

Reviewer 2:

(1)

While this does appear to be a good candidate variant, there are numerous instances where the language describing the potential role is not appropriate based on the analyses yet completed. For example lines 34-35 (“…strongly suggests…caused”), line 63 (“…due to a 3 base pair…”), and line 276 (“This strongly suggest…”). The authors should rephrase these (and any other) passages to reflect potential association as opposed to causation.

Response: We applied the ACMG/AMG consensus standards for the interpretation of sequence variants in human patients (Richards et al. 2015 Genet Med 17:405-424) and extrapolated the human guidelines to dogs. Using this approach we have the following support for the causality of the KRT:p.Asn190del variant:

• PS2, de novo in a patient with the disease and no family history

• PM2, mutant allele is absent from controls

• PM4, protein length change due to in-frame deletion in a non-repeat region

• PM5, novel missense change (in our case a single amino acid deletion) where a different missense change determined to be pathogenic has been seen before

• PP4, patient's phenotype is highly specific for a disease with a single genetic etiology

Thus, we have 1 strong, 3 moderate and 1 supporting criteria that support the causality. According to human diagnostic standards, this is sufficient to classify the variant as pathogenic. We therefore think that our evidence is sufficient to claim causation. We chose a strong ("strongly suggests to cause") rather than an absolute ("causes") wording to take into account the cross-species extrapolation. We added a paragraph at the end of the discussion outlining this argumentation.

(2)

Line 31 (and the discussion) speculate that the deletion “interferes with keratin dimerization”. While this certainly could be true based on the previous literature, this has not been thoroughly investigated here. The immunohistochemistry presented herein does not suggest this really. If the authors want to include a description of this potential mechanism the language should be toned down (as in the first comment) and likely removed from the abstract. Alternatively, and if possible, it would be really interesting to see the results from some type of in-vitro protein-protein interaction assay.

Response: We agree with the reviewer that our manuscript lacks experimental support for the proposed defect in keratin dimerization. We modified the statement in the abstract and the discussion, which now reads: "This might indicate that the variant possibly interferes with keratin dimerization or another function of KRT1." We hope that it is acceptable to keep the proposed hypothetical pathomechanism in the abstract.

(3)

How was the variant filtering accomplished with the 793 public control genomes? In addition, I noticed there were no Shar Pei’s in the control set. Were there none available or were they excluded? A reference (14) was used to cover how the mapping and variant calling were conducted but additional descriptions of how the variant filtering was conducted, why these 793 were selected, and why no other healthy Shar Pei’s were included is needed.

Response: We added a description of our variant filtering process to the methods. The 793 control genomes represented a convenience sample that we also used for other similar projects in the past. It is an expanded version of the DBVDC dataset described in the reference Jagannathan et al. 2019. Preparing and handling such massive datasets with hundreds of mammalian genomes (~50 Tb raw sequencing data) takes many weeks of computing time at the high performance computing cluster of the University of Bern. While the inclusion of non-affected unrelated Shar Pei controls would have been desirable from a scientific point of view, the costs for the generation and analysis of such data would have been prohibitive for our study.

(4)

Why was confirmation of parentage done with VCFtools as opposed to a more commonly used parentage/genetic marker test? In this section they are referred to as “presumed parents” but “unaffected parents” earlier suggesting not presumed. Similarly, lines 211-212 state parentage was “confirmed based on the genome sequence data.” Please include clarification as to the why and how here – this is important so that we have absolute confidence that the parents are in fact the parents.

Response: We repeated the parentage confirmation and revised the methods section accordingly. There was no need to perform additional laboratory experiments as we had the full genome sequences of the trio consisting of the offspring and both parents. We extracted the genotypes at more than 6 million informative variants to confirm the parentage, which is "even better" than a routine parentage test that would be based on either 10-20 microsatellite markers or less than 300 single nucleotide variants. We hope that the revised methods section is now sufficiently detailed to clear all possible doubts about the true parentage.

(5)

Line 122 the authors should include a description of the multiple sequence alignment method.

Response: We added the requested method (reference to NCBI HomoloGene website).

(6)

The candidate gene set appears to be from three previous papers, one of which is for a dog with ichthyosis. Were there attempts to expand the search for candidate genes via databases such as OMIM or PubMed? Additional descriptions on why only these sources were used should be included. Perhaps a structured systematic review is not needed but a more complete explanation on how the candidate gene set was developed would be helpful in understanding why this severe filtering (92k to 500 variants) was needed.

Response: We think that it is a standard approach in medical genetics (for humans and animals) to focus on known functional candidate genes when single cases with presumed inherited diseases with characteristic phenotypes such as an ichthyosis are investigated. For our list of candidate genes, we compiled the information from two human review articles listing a total of 35 functional candidate genes. These two reviews are based on systematic reviews of OMIM and the scientific literature. We added ASPRV1 as this is an ichthyosis gene that was only recently discovered in dogs and subsequently in human ichthyosis patients and is not yet contained in the older reviews.

As our initial hypothesis-based functional candidate gene approach apparently worked, we don't think it is appropriate to provide a lengthy verbose justification of the selection of candidate genes. Such lists are always debatable. If the initial approach had been unsuccessful, we would of course have performed additional and possibly hypothesis-free analyses.

We revised the S3 table and now provide a full list of all private variants in the affected dog. Thus, interested readers have the opportunity to reproduce our analyses and/or to perform the filtering and prioritization steps of the ~7 million starting variants in different orders. We are convinced that, independent on the specific order of filtering steps, the KRT1 variant will always be identified as the most likely cause of the observed phenotype.

(7)

Table 1 lists 92k private variants, were there any attempts to examine this set for other potential variants compatible with an autosomal recessive inheritance pattern that could explain the phenotype? The definition of “protein-changing” should be probably be included somewhere as well. I would suggest including some type of analyses on the 92k private variants. For example, theses could be categorized by location, type, and predicted impact (I believe SnpEff outputs summary files containing this type of data?). Similar to the candidate gene comment above, the authors should examine at least the predicted high-impact variants for other potential genes that may be associated with keratinization disorder.

Response: We revised and expanded the methods and results section. When the genotypes of the parents and a protein-changing effect (SnpEff impact high or moderate) are taken into account, of the starting 92 k private variants only 46 remain and only 1 of those is in a functional candidate gene.

(8)

Lines 231 and 235 state expression was “comparable” between the affected and normal dogs. It looks a bit a subjective and to the untrained eye, these appear rather different on paper. It would seem that in order to make statements about comparability, protein quantification should be conducted. Transcript quantification via RNA-seq or qPCR may be warranted as well to uncover any potential differences in expression. Baring any additional experiments, perhaps there is a pathologist associated with this study that could describe how these were identified as comparable?

Response: Histology and immunohistochemistry has been reviewed by a board-certified veterinary pathologist specialized in dermatopathology and experienced in evaluating immunohistochemical stains. We revised the text section describing the results of the immunohistochemistry experiment.

(9)

There should probably be some mention in the discussion of the uncertainty inherent in using the public annotation as there are known issues. For example, does this deletion appear in the same amino acid and domain using the Ensembl annotation which lists two isoforms?

Response: We added a verbose statement on the limitations of our approach to the discussion. We also added a comment and a reference that the NCBI annotation that we used for the study corresponds to a manually curated annotation for the canine KRT1 gene. This is another example, where the NCBI annotation seems to be superior to the ENSEMBL annotation for the dog. We consider the ENSEMBL annotation for the canine KRT1 gene incorrect.

(10)

There is a discrepancy in author order between the cover sheet and manuscript. Please correct.

Response: the order of listed Authors is the same on cover letter as well as manuscript 

(11)

Lines 46-57 is a single sentence and as a result a little difficult to follow. I would suggest breaking this up or restructuring it.

Response: We revised this long enumeration and built smaller sentences.

(12)

Table 1 legend “genome resequencing”. Is this supposed to be “genome sequencing” or was this dog previously sequenced?

Response: We used the term "resequencing" to indicate that we did not perform a de novo genome assembly for this dog, but rather aligned the short-read sequence data to an existing reference genome. While this is the technically correct term, we realize that it may be confusing to the non-specialist reader. Therefore, we now replaced it with "whole genome sequencing".

(14)

Figure 3 legend has a reference error {Bray, 2015 #17}.

Response: Revised accordingly.

(15)

Minor inconsistency in naming the protein dimer (line - 266 KRT1/10; line 273 - KRT1/KRT10). I am not sure which is correct but should be consistent I would think.

Response: We consistently revised the entire manuscript and use now KRT1 instead of keratin 1 or K1.

---

## [Decision Letter · Decision Letter 1]

15 Sep 2022

A de novo variant in the keratin 1 gene (KRT1) in a Chinese shar pei dog with severe congenital cornification disorder and non-epidermolytic ichthyosis

PONE-D-21-30440R1

Dear Dr. Affolter,

We’re pleased to inform you that your manuscript has been judged scientifically suitable for publication and will be formally accepted for publication once it meets all outstanding technical requirements.

Kind regards,

Gerhard Wiche, Ph.D.

Academic Editor

PLOS ONE

Additional Editor Comments (optional):

Reviewers' comments:

Reviewer's Responses to Questions

**Comments to the Author**

1. If the authors have adequately addressed your comments raised in a previous round of review and you feel that this manuscript is now acceptable for publication, you may indicate that here to bypass the “Comments to the Author” section, enter your conflict of interest statement in the “Confidential to Editor” section, and submit your "Accept" recommendation.

Reviewer #1: All comments have been addressed

Reviewer #2: All comments have been addressed

2. Is the manuscript technically sound, and do the data support the conclusions?

Reviewer #1: Yes

Reviewer #2: Yes

3. Has the statistical analysis been performed appropriately and rigorously? 

Reviewer #1: Yes

Reviewer #2: N/A

4. Have the authors made all data underlying the findings in their manuscript fully available?

Reviewer #1: Yes

Reviewer #2: Yes

5. Is the manuscript presented in an intelligible fashion and written in standard English?

Reviewer #1: Yes

Reviewer #2: Yes

6. Review Comments to the Author

Reviewer #1: The authors have adequately addressed all of the points that I raised in the initial review. I am comfortable now recommending this manuscript for publication.

Reviewer #2: All comments/questions were addressed thoughtfully by the authors. The manuscript is technically sound and the underlying appears to be publicly available.

7. PLOS authors have the option to publish the peer review history of their article (what does this mean?). If published, this will include your full peer review and any attached files.

Reviewer #1: No

Reviewer #2: No

---

## [Editor Report · Acceptance letter]

27 Sep 2022

PONE-D-21-30440R1 

A *de novo* variant in the keratin 1 gene (*KRT1*) in a Chinese shar-pei dog with severe congenital cornification disorder and non-epidermolytic ichthyosis 

Dear Dr. Affolter:

I'm pleased to inform you that your manuscript has been deemed suitable for publication in PLOS ONE. Congratulations! Your manuscript is now with our production department. 

Kind regards, 

on behalf of

Prof. Gerhard Wiche 

Academic Editor

PLOS ONE